# Asymmetric benzylic C(sp³)−H acylation via dual nickel and photoredox catalysis

Leitao Huan[1,2], Xiaomin Shu [1,2], Weisai Zu[1], De Zhong[1] & Haohua Huo [1✉]

Asymmetric C(sp³)−H functionalization is a persistent challenge in organic synthesis. Here, we report an asymmetric benzylic C−H acylation of alkylarenes employing carboxylic acids as acyl surrogates for the synthesis of α-aryl ketones via nickel and photoredox dual catalysis. This mild yet straightforward protocol transforms a diverse array of feedstock carboxylic acids and simple alkyl benzenes into highly valuable α-aryl ketones with high enantioselectivities. The utility of this method is showcased in the gram-scale synthesis and late-stage modification of medicinally relevant molecules. Mechanistic studies suggest a photocatalytically generated bromine radical can perform benzylic C−H cleavage to activate alkylarenes as nucleophilic coupling partners which can then engage in a nickel-catalyzed asymmetric acyl cross-coupling reaction. This bromine-radical-mediated C−H activation strategy can be also applied to the enantioselective coupling of alkylarenes with chloroformate for the synthesis of chiral α-aryl esters.

[1] State Key Laboratory of Physical Chemistry of Solid Surfaces, Key Laboratory of Chemical Biology of Fujian Province, College of Chemistry and Chemical Engineering, Xiamen University, Xiamen, People's Republic of China. [2] These authors contributed equally: Leitao Huan, Xiaomin Shu. ✉email: hhuo@xmu.edu.cn

Chiral α-aryl ketones are versatile building blocks and represent important pharmacophores existing in many drug molecules such as ibuprofen and naproxen[1,2]. Although numerous enantioselective approaches for preparing quaternary α-aryl ketones have been reported[3–5], asymmetric methods to access more commonly encountered tertiary variants remain limited presumably owing to the lability of tertiary stereocenters[6]. Nevertheless, transition-metal catalyzed asymmetric couplings of aryl organometallic reagents with α-bromo ketones[7–9], benzylic zinc reagents with thioesters[10], benzylic chlorides with acid chlorides under reductive conditons[11], and aryl alkenes with activated carboxylic acids in the presence of a hydrosilane[12,13] have been disclosed in seminal studies by Fu, Maulide, Reisman, and Buchwald, respectively (Fig. 1a). Despite this impressive progress, it remains highly desirable to develop complementary methods that use feedstock functional groups to avoid sensitive organometallic reagents, preformed organohalides, and stoichiometric reductants.

In recent years, nickel and photoredox dual catalysis have emerged as a powerful tool for the direct C(sp³)−H functionalization of feedstock hydrocarbons by leveraging photoredox-mediated C−H activation and nickel's unique ability in alkyl cross-couplings[14–33] (Fig. 1b). This strategy allows the use of mild and robust conditions to perform C−H cleavage via a hydrogen atom transfer (HAT) or single-electron transfer (SET) pathway. These routes provide attractive strategic alternatives to challenging metal-catalyzed C(sp³)−H functionalization without the need for high reaction temperature and coordinating directing groups that are often encountered in traditional C−H activation reactions[34]. Pioneering works by MacMillan, Molander, and Doyle have led to numerous nonasymmetric methods for dual nickel/photoredox catalyzed C(sp³)−H coupling reactions[17–33]. In contrast, enantioselective approaches remain largely undeveloped[35]. Few successful examples in nickel and photoredox catalyzed asymmetric C(sp³)−H functionalization are only limited to the arylation of C(sp³)−H bonds with aryl bromides[36,37].

Recently, our laboratory reported a direct enantioselective C(sp³)−H acylation of N-alkyl benzamides for the synthesis of α-amino ketones; wherein, a chiral nickel catalyst could engage photocatalytically generated α-amino radicals and in situ-activated carboxylic acids in acyl cross-couplings[38]. We reasoned that this strategy could be applied to the asymmetric benzylic C−H acylation of alkylarenes to address the challenges described above for the synthesis of α-aryl ketones via radical C(sp³)−H functionalization[36–38]. Despite that initial progress[38], no examples of enantioselective benzylic C−H acylation have been reported. In addition, there is an increasing demand for the development of benzylic C−H functionalization reactions for the synthesis of high value-added molecules from simple alkylarenes[39–50]. In this work, we report an enantioselective benzylic C−H acylation of alkylarenes with in situ-activated carboxylic acids enabled by nickel and photoredox dual catalysis (Fig. 1c, top).

## Results

**Reaction design**. The proposed catalytic cycle for this benzylic acylation is shown in the bottom of Fig. 1c. It has been reported that single-electron oxidation of bromide anion by photoexcited photocatalyst can generate bromine radical ($E_{1/2}[Ir(III^*/II)] = +1.21$ V vs SCE in CH₃CN; $E_{1/2}^{ox}$ [Br⁻/Br·] = +0.80 V vs SCE in DME)[51–54]. According to the literature precedent and our previous mechanistic experiments[38,51–54], we hypothesize that the catalytic reaction is initiated by oxidative addition of Ni(0) catalyst **I** to an in situ-activated carboxylic acid to afford Ni(II) species **II**. Subsequent trapping of prochiral benzylic radicals generated from the bromine-radical-mediated HAT process provides Ni(III) complex **III**, which undergoes reductive elimination to yield the desired product and Ni(I) species **IV**. A recent computational study of nickel-catalyzed cross-coupling of photoredox-generated benzylic radicals suggested that reductive elimination is the stereochemistry-determining step[55]. Finally, SET between Ni(I) species **IV** and reduced photocatalyst regenerates the Ni(0) catalyst **I** and ground-state photocatalyst to close

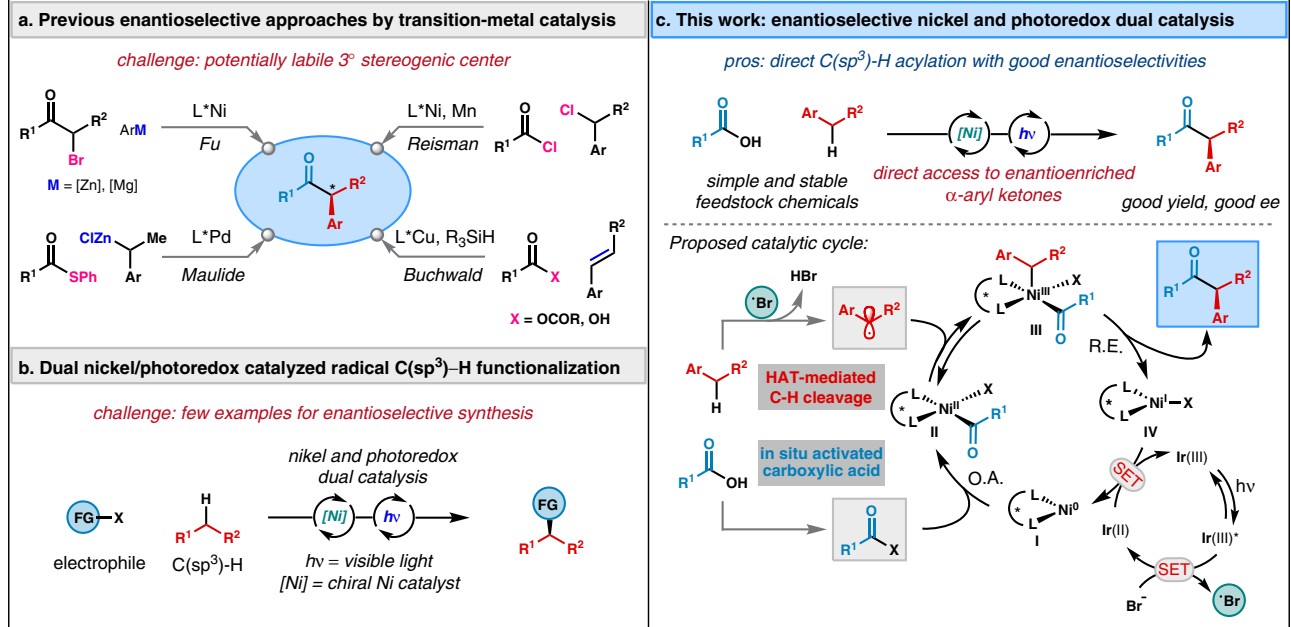

**Fig. 1 Enantioselective metal-catalyzed approaches for the synthesis of α-aryl ketones. a** Previous approaches. **b** Dual nickel/photoredox catalyzed C(sp³)−H functionalization. **c** This work and its mechanistic hypothesis. *O. A.* oxidative addition, *R. E.* reductive elimination. Ir(III) = Ir[dF(CF₃)ppy]₂(dtbbpy)PF₆.

**Table 1 Effect of reaction parameters.**

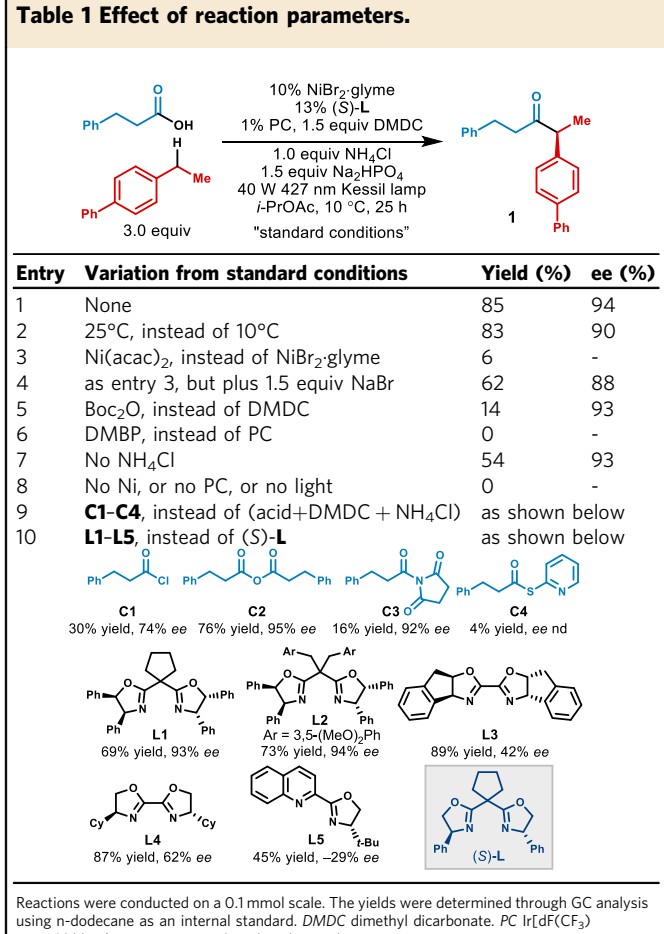

| Entry | Variation from standard conditions | Yield (%) | ee (%) |
|---|---|---|---|
| 1 | None | 85 | 94 |
| 2 | 25°C, instead of 10°C | 83 | 90 |
| 3 | Ni(acac)$_2$, instead of NiBr$_2$·glyme | 6 | - |
| 4 | as entry 3, but plus 1.5 equiv NaBr | 62 | 88 |
| 5 | Boc$_2$O, instead of DMDC | 14 | 93 |
| 6 | DMBP, instead of PC | 0 | - |
| 7 | No NH$_4$Cl | 54 | 93 |
| 8 | No Ni, or no PC, or no light | 0 | - |
| 9 | **C1–C4**, instead of (acid+DMDC + NH$_4$Cl) | as shown below | |
| 10 | **L1–L5**, instead of (S)-**L** | as shown below | |

Reactions were conducted on a 0.1 mmol scale. The yields were determined through GC analysis using n-dodecane as an internal standard. DMDC dimethyl dicarbonate. PC Ir[dF(CF$_3$) ppy]$_2$(dtbbpy)PF$_6$, DMBP 4,4'-dimethoxybenzophenone.

both catalytic cycles ($E_{1/2}^{red}$ [Ir(II/III)] = −1.37 V vs SCE in CH$_3$CN).

**Reaction optimization.** Our investigation began with an exploration of reaction conditions for the coupling of 4-ethylbiphenyl and 3-phenylpropanoic acid (Table 1). Based on previously reported elegant strategies and our recent conditions for carboxylic acid activation in ketone synthesis[38,56–60], dimethyl dicarbonate (DMDC) was chosen as the activating agent to generate mixed anhydride in situ from carboxylic acids. After an extensive study of reaction parameters (also see Supplementary Table 1), we were delighted to find that a simple chiral nickel/bis (oxazoline) catalyst and a known Ir-photocatalyst could provide the acylation product in 85% yield and 94% ee (entry 1). An attractive feature of this transformation is that only commodity chemicals are involved in this reaction. From the standpoint of commercial availability, carboxylic acids are perhaps the most ubiquitous functional group. The reaction could be also performed at room temperature with similar efficiency (entry 2). The use of a nickel source free of bromide led to almost no product formation (entry 3). Interestingly, the addition of NaBr was found to restore the reaction with comparable outcome, showcasing the crucial role of bromide anion in the catalytic cycle (entry 4). The use of Boc$_2$O instead of DMDC provided the desired product in 93% ee, but with poor yield (entry 5). Replacing the Ir-photocatalyst with a ketone triplet sensitizer, which has been employed in nickel/photoredox catalyzed C(sp$^3$)−H functionalization reactions[18,23,36] resulted in no product formation (entry 6). Running the reaction in the absence of NH$_4$Cl, which has been

previously employed to facilitate the formation of a mixed anhydride[56,60], led to a significantly lower yield (entry 7). Control experiments revealed nickel, photocatalyst, and light are indispensable for product formation (entry 8). The use of other acyl surrogates in place of the in situ combination of carboxylic acid, DMDC, and NH$_4$Cl did not provide improvements (entry 9). Other chiral ligands such as **L1** and **L2** delivered acylation products with similar enantioselectivities, albeit in diminished yields (entry 10).

**Evaluation of substrate scope.** We next investigated the scope for cross-coupling of alkylarenes with carboxylic acids employing the optimized reaction conditions (Fig. 2). This transformation was compatible with many functional groups, such as chloride (**7**, **44**), bromide (**8**, **43**, **32**, and **33**), fluoride (**9**, **19**, **24**, and **31**), ether (**10**, **14**, and **42**), nitrile (**11**), carbamate (**12**), ester (**13**, **40**, and **41**), olefin (**15**, **16**), boronate ester (**34**), pyrazole (**35**), and heteroaromatic moieties (**17**, **36**, and **46**). Remarkably, the alkyl halide, aryl halide, aryl boronate ester, and terminal olefin can serve as versatile synthetic handles for further structural elaborations. Pyrazole- and thiophene-based heterocycles are commonly found in pharmaceutically relevant compounds. The coupling of 4-ethylbiphenyl with carboxylic acids bearing different steric properties furnished the corresponding α-aryl ketones in high yields and ee's (**2**−**5**). Aromatic carboxylic acids were also suitable coupling partners to generate desired products, albeit with modest yields and enantioselectivities (**18**, **19**). The corresponding methyl carboxylate was a significant side product for the cross-coupling of aromatic carboxylic acids. For the alkylarene component, acylation of para-substituted alkylarene bearing diverse electronic properties resulted in good yields and enantioselectivities (**20**−**27**). When the alkylarene featuring more than one benzylic C−H site was used, monoacylation products could be obtained in good yields and ee's (**28**−**30**). The homobenzylic bulky substrate was a competent coupling partner (**39**). Acylation of indane provided **45** in good yield and slightly reduced enantioselectivity. Under the current reaction conditions, the sterically hindered coupling partners such as the α-branched carboxylic acids and ortho-substituted alkylarene (**33**) led to low efficiency or no product formation (also see Supplementary Table 1).

**Late-stage functionalization.** Given the particularly broad functional group tolerance of our method, we sought to demonstrate the utility of this operationally convenient method in the late-stage functionalization of medicinally relevant molecules (Fig. 3)[61]. Specifically, acylation of benzylic C−H bonds of drugs such as ibuprofen, fenoprofen, ketoprofen, and naproxen, provided corresponding drug analogs in good yields and enantioselectivities (**47**−**51**). Employing menthol and amino acid derivatives as alkylbenzene coupling partners led to good diastereoselectivities (**52**−**55**). With oxaprozin, stearic acid, oleic acid, 2,4-D, and lithocholic acid derivatives as acyl donors, the acylation proceeded with good stereoselectivity (**56**−**61**).

**Gram-scale synthesis and parallel synthesis.** To demonstrate the scalability of the present method, two 20.0 mmol scale reactions were performed in a common flask to produce 5.35 g of chiral ketone product **8**, and 9.13 g of lithocholic acid derivative **60** with excellent stereoselectivity and good yield (Fig. 4a). To further demonstrate the synthetic utility, two types of drug analogs derived from (S)-flurbiprofen and artesunate were prepared in parallel with high yields and excellent stereoselectivities (Fig. 4b). More than 100 mg of product was obtained in all cases. It is noteworthy that the labile peroxide subunit in artesunate was tolerated particularly well under mild conditions. This powerful

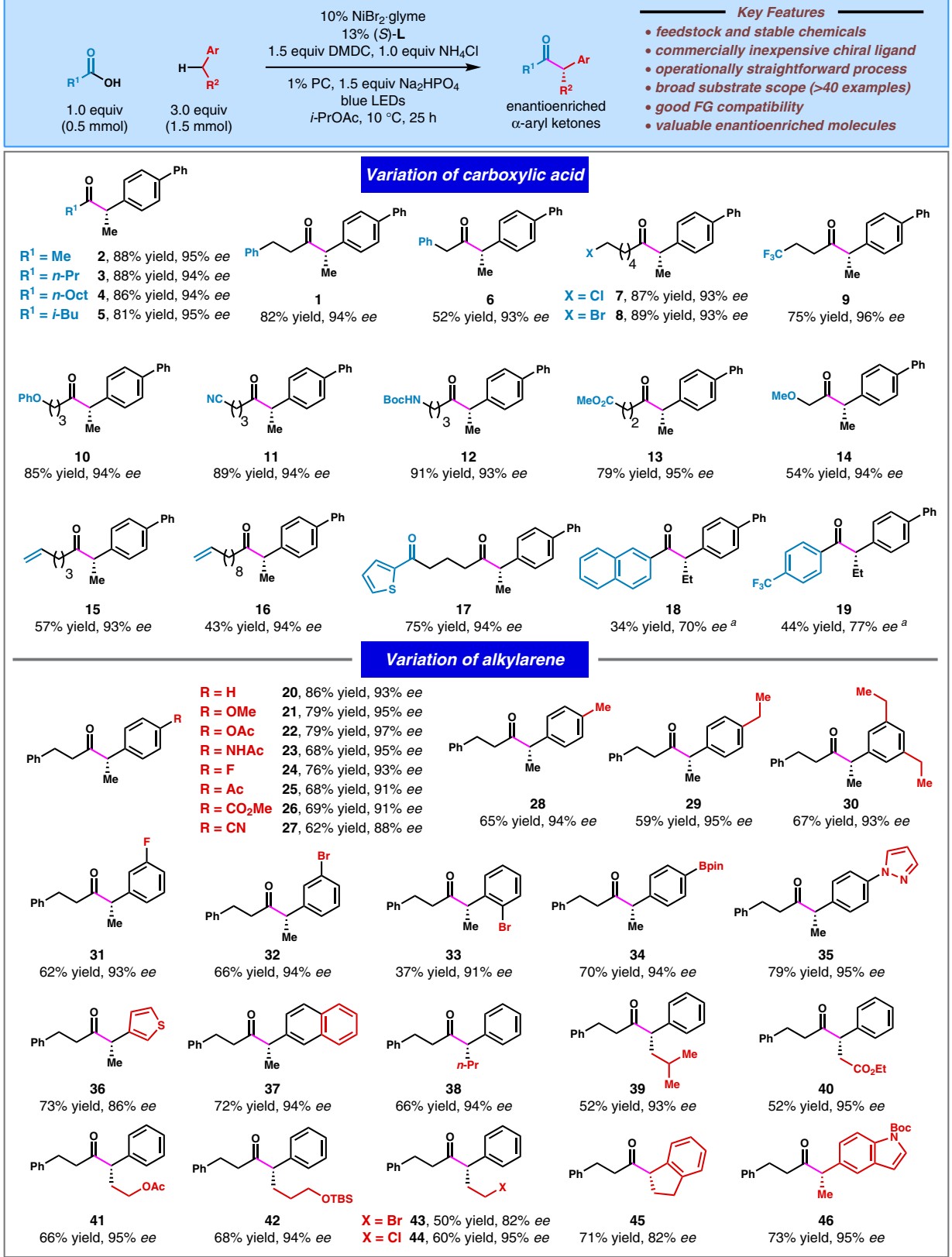

**Fig. 2 Substrate scope of enantioselective acylation of benzylic C(sp³)—H bonds with carboxylic acids.** All data represent the average of two experiments. Unless otherwise stated, reactions were conducted on a 0.5 mmol scale under standard conditions. [a]In place of the standard conditions, chiral ligand **L3**, 3.0 equiv DMDC, and 3.0 equiv $K_2HPO_4$ were used.

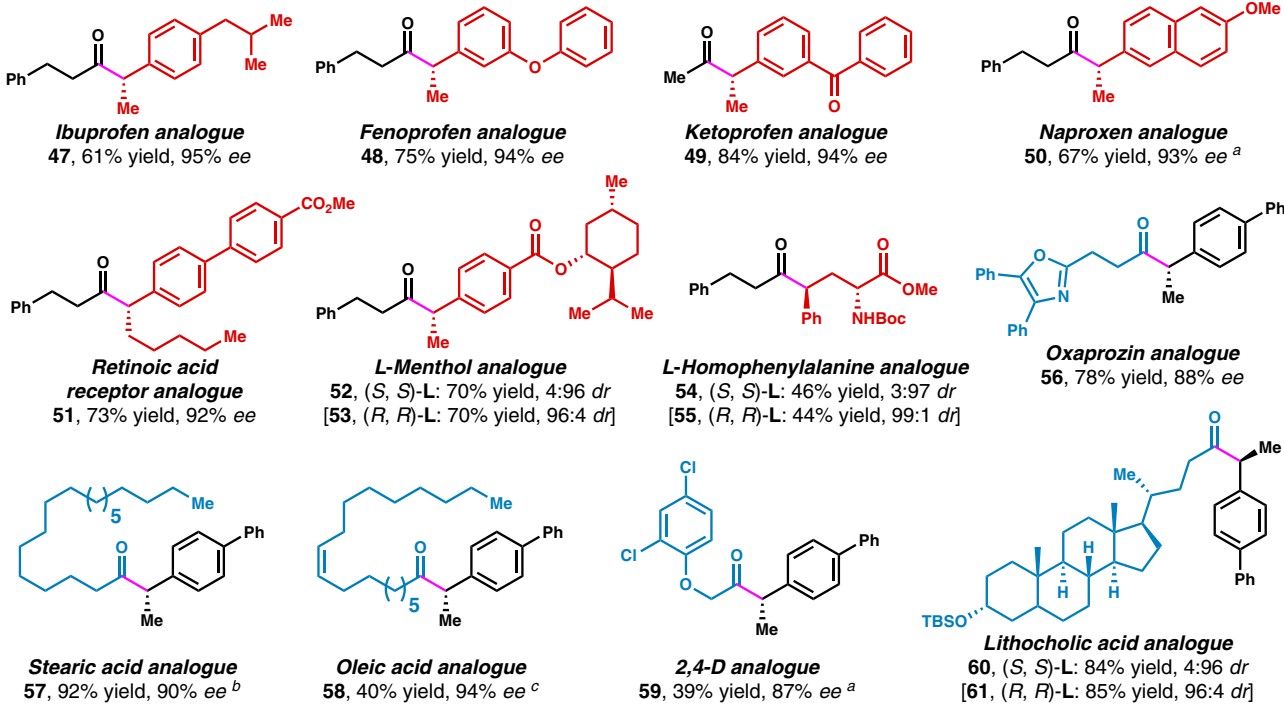

**Fig. 3 Late-stage functionalization.** All data represent the average of two experiments. Unless otherwise stated, reactions were conducted on a 0.5 mmol scale under standard conditions. [a]In place of the standard conditions, the reaction was conducted at 25 °C in dioxane. [b]In place of the standard conditions, the reaction was conducted at 25 °C. [c]In place of the standard conditions, 5.0 equiv of 4-ethylbiphenyl was used.

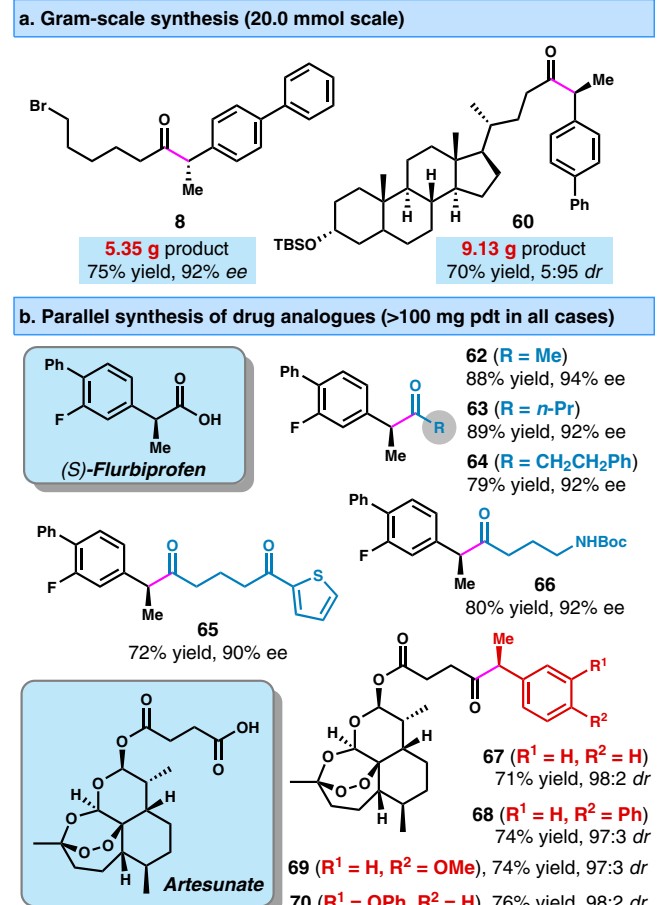

**Fig. 4 Gram-scale synthesis and parallel synthesis. a** Gram-scale synthesis. **b** Parallel synthesis of drug analogs.

method enables the streamlined synthesis of drug analogs, providing attractive opportunities for the rapid exploration of structure-activity relationships in drug discovery[62], as well as complementing the existing methods for the synthesis chiral α-aryl ketones[7–13].

**Mechanistic observations**. We next performed preliminary mechanistic studies for this newly developed method (Fig. 5). The primary kinetic isotope effect was observed in intermolecular parallel and competition experiments, which suggested that C−H cleavage significantly contributed to the rate-determining step (Fig. 5a). When the reaction was performed in the presence of an electron-deficient alkene, the benzylic acylation was completely inhibited, and a racemic adduct **71** was obtained in 58% yield (Fig. 5b, top). This observation supported the benzylic radical might be involved in the catalytic cycle. Moreover, in the absence of nickel catalyst and in situ-generated acyl electrophile (Fig. 5b, bottom), the addition of 1.5 equiv of NaBr to the coupling of 4-ethylbiphenyl with electron-deficient alkene led to the adduct **71** in 16% yield, which suggested photochemical oxidatively generated bromine radical was likely involved in the acylation reaction.

**Rational expansion**. Finally, we questioned whether this bromine-radical-mediated C−H cleavage strategy could be applied to the synthesis of α-aryl esters rather than α-aryl ketones[63–65]. Indeed, replacing the in situ-generated mixed anhydride with commercially available phenyl chloroformate led to a number of α-aryl esters in good yields and selectivities under similar conditions (Fig. 6). Chiral ligand **L2** proved to be optimal for this transformation.

**Discussion**

In summary, a direct enantioselective benzylic C(sp³)−H acylation for the synthesis of α-aryl ketones has been developed. Several attractive features are noteworthy. First, both coupling

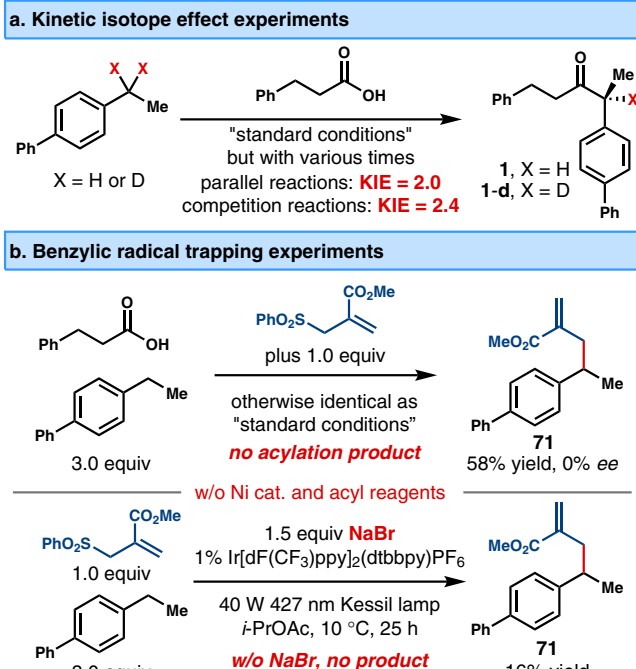

**a. Kinetic isotope effect experiments**

"standard conditions"
but with various times
parallel reactions: **KIE = 2.0**
competition reactions: **KIE = 2.4**

X = H or D

**1**, X = H
**1-d**, X = D

**b. Benzylic radical trapping experiments**

plus 1.0 equiv

otherwise identical as
"standard conditions"
***no acylation product***

3.0 equiv

**71**
58% yield, 0% *ee*

w/o Ni cat. and acyl reagents

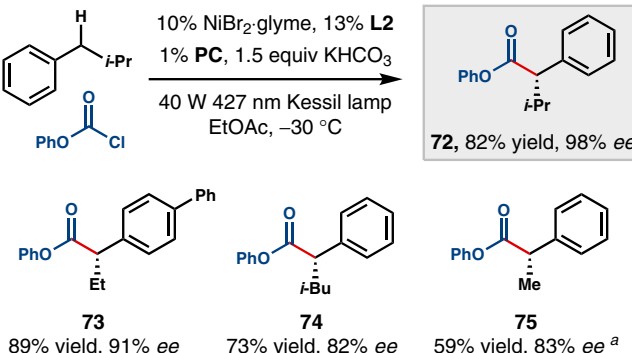

1.5 equiv **NaBr**
1% Ir[dF(CF₃)ppy]₂(dtbbpy)PF₆

40 W 427 nm Kessil lamp
*i*-PrOAc, 10 °C, 25 h
***w/o NaBr, no product***

1.0 equiv

3.0 equiv

**71**
16% yield

**Fig. 5 Mechanistic observations. a** Kinetic isotope effect experiments. **b** Benzylic alkyl radical trapping experiments.

10% NiBr₂·glyme, 13% **L2**
1% **PC**, 1.5 equiv KHCO₃

40 W 427 nm Kessil lamp
EtOAc, −30 °C

**72**, 82% yield, 98% *ee*

**73**
89% yield, 91% *ee*

**74**
73% yield, 82% *ee*

**75**
59% yield, 83% *ee* [a]

**Fig. 6 Rational expansion for the synthesis of α-aryl esters.** All data represent the average of two experiments. Unless otherwise noted, reactions were conducted on a 0.5 mmol scale under stated conditions. [a]In place of the stated conditions, the reaction was conducted at −40°C with 5.0 equiv ethylbenzene.

partners, carboxylic acids and alkylbenzenes, have broad commercial availability. Second, this operationally simple and scalable method has a broad substrate scope and excellent functional group tolerance. Third, this mild protocol can be applied to the late-stage modification of pharmaceutically relevant molecules. Finally, the asymmetric synthesis of α-aryl esters is also accessible based on a simply rational expansion. The development of enantioselective C(sp³)−H alkylation for the construction of C(sp³)−C(sp³) bonds is underway in our laboratory.

## Methods

**Representative procedure for the synthesis of α-aryl ketone 1.** In a nitrogen-filled glovebox, Ir[dF(CF₃)ppy]₂(dtbbpy)PF₆ (5.5 mg, 0.005 mmol), NiBr₂·glyme (15.5 mg, 0.050 mmol), (S)-**L** (23.5 mg, 0.065 mmol), NH₄Cl (26.5 mg, 0.50 mmol), Na₂HPO₄ (106.5 mg, 0.75 mmol), a Teflon stir bar, and anhydrous *i*-PrOAc (5.0 mL) were added sequentially to a 15 mL vial. The reaction mixture was stirred at

room temperature for 30 min, after which it turned to a purple suspension. Next, 3-phenylpropanoic acid (75.0 mg, 0.50 mmol) was added as a solid, followed by addition of 4-ethylbiphenyl (0.75 mL, 2.0 M in *i*-PrOAc, 1.50 mmol) via a 1.0 mL syringe. The vial was then capped with a polytetrafluoroethylene septum cap, and DMDC (80.0 µL, 0.75 mmol) was added via a 100 µL syringe. The vial was next transferred out of the glovebox, and vacuum grease was applied to cover the entire top of the septum cap. Then, the reaction mixture was stirred at 10 °C in an EtOH bath for 5 min, followed by irradiation with a 40 W blue LED lamp (Kessil PR160L, 427 nm). The reaction was stirred at 10 °C under irradiation for 25 h. The reaction mixture was then passed through a short pad of silica gel, with Et₂O as the eluent (~35 mL). The resulting mixture was concentrated, and the residue was purified by flash chromatography on silica gel, which provided the desired acylation product **1** in 82% yield and 94% ee as a white solid. All new compounds were fully characterized (See the Supplementary Methods).

## Data availability

The data that support the findings of this study are available within the article and its Supplementary Information files. The X-ray crystallographic coordinates for structures reported in this article have been deposited at the Cambridge Crystallographic Data Centre (CCDC), under deposition number CCDC 2058381 (**59**). The data can be obtained free of charge from The Cambridge Crystallographic Data Centre via http://www.ccdc.cam.ac.uk/data_request/cif.

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

## Acknowledgements

This work is dedicated to 100th anniversary of Xiamen University and its Chemistry. We are grateful for financial support from the NSFC (22071203) and the Xiamen University. We thank professor Hai-Chao Xu (Xiamen University) for assistance and helpful discussions. We acknowledge Wesley Harrison (UIUC) for manuscript revision.

## Author contributions

L.H. and X.S. contributed equally to this work. L.H., X.S., W.Z., and D.Z. performed the experiments and analyzed the data. H.H. designed and directed the project and wrote the manuscript.

## Competing interests

The authors declare no competing interests.

**Additional information**

