## [Peer Review File · Nature Communications]

REVIEWER COMMENTS

Reviewer #1 (Remarks to the Author):

The authors report benzylic C-H acylation via dual photoredox/nickel catalysis. This reviewer has some mixed feelings about the appropriateness of this submission for publication in Nature Communications. On the one hand, the method developed looks very robust and versatile, and a high level of rigor and attention to detail is exhibited. On the other hand, the method is somewhat derivative of previous research, and therefore conceptually not much is added to the literature other than the report of a facile route to a class of molecules that are, admittedly, important.

Detailed points to be considered:

- 1) In places there is a need for an English-speaking reader to correct for idiomatic English.
- 2) There is not a great deal of attention paid to previous asymmetric processes through photoredox catalysis. The authors are referred to a recent review: *Angew. Chem. Int. Ed.* 2021, 60, 1714 – 1726.
- 3) There is also short-shrift paid to previous acylation reactions conducted by in situ activation of carboxylic acids. See, for example, *Org. Lett.* 2017, 19, 3612–3615 and references therein. There are several other examples since this, but this is one of the earliest examples using dimethyl dicarbonate.
- 4) It is not clear what differentiates acetate and esters (lines 94-95). They are both esters.
- 5) In lines 35 and 36, the compounds are not boranes, but rather boronates.
- 6) The chemistry in Figure 4B with Flurbiprofen is not well illustrated – it is not clear from the graphics exactly what is being done. The wrong bond disconnects are highlighted and the colors are not consistent with those used in the rest of the manuscript. This must be corrected for clarity.
- 7) In Figure 5, it is surprising that addition to the alkene was competitive with capture of the radical by Ni complex, a process that is essentially barrierless. Perhaps this is just the effect of concentration in that there is a large excess of the alkene in contrast to the catalytic Ni complexes.
- 8) A glaring omission is that the proposed mechanism does not address the source of asymmetric induction. Thus, the enantioselectivity could result from the irreversible diastereoselective formation of the Ni complex or a dynamic kinetic transformation with reversible addition of the radical to the nickel complex. There are calculations from other groups that address this issue, but no discussion is evident in this contribution.
- 9) It is not clear from the experimental description how the reactions were kept at 10 °C during irradiation. This should be explicitly documented.
- 10) There are no melting points reported for any of the solids.

Overall a very solid contribution, but perhaps lacking a bit in novelty and scholarship.

Reviewer #2 (Remarks to the Author):

In this manuscript, Huan and coworkers report a photoredox/Ni/HAT promoted asymmetric acylation of benzylic C-H bonds to access α -aryl ketones with broad substrate scope and functional group tolerance. While the photoredox/Ni catalytic platform has been well established, the development of asymmetric transformations remains less explored. Although this paper is a logical extension of their previously reported work (ref. 37) using alkyl arenes instead of amide as C-H coupling partner under very similar conditions, the synthesis of chiral α -aryl ketone in the manuscript exhibits meaningful complementarity (simple use of acids as acyl source and alkyl arenes as prochiral C-H nucleophiles) compared with previous enantioselective approaches by transition-metal catalysis, which is especially well demonstrated in Fig. 3. Overall, this is a well done, very nice study that should be accepted after some revisions noted below.

Prior to publication, I recommend the following points be addressed:

1. The authors demonstrate aromatic and aliphatic (primary) acids as partners. Some comment would be useful with respect to the use of secondary and tertiary acids?
2. Why do aromatic acids dramatically lower reactivity and enantioselectivity? A comment would

be useful.

3. Argument of line 106 is specious. The capability of a method is not justified by whether the product has been previously prepared or not. Clearly, if the methyl substrate is known, one can make the ethyl derivative and claim new chemical space. This should be deleted. Also, in line 106, 'figure 3' should be 'figure 2'.

4. The colours in Fig. 3 B make it confusing for the disconnections of coupling partners. Sometimes the blue is acid, sometimes not, sometimes it includes the carbonyl, sometimes not. This should be fixed.

5. A recent publication about partial activation of alcohols in cross electrophile coupling reveals an intriguing concept (JACS2021/143/513). A simple experiment can verify whether the in-situ acid activation follows this kinetics.

6. After acylation, the benzylic position in the product is deactivated. In principle, less activated C-H precursors should also work.

7. The paper (JACS2015/137/4896) discusses stereoconvergence in Photoredox/Ni-catalyzed cross coupling involving benzylic radical capture and should be cited.

Reviewer #3 (Remarks to the Author):

The manuscript by Huo and coworkers reports a new method for the asymmetric benzylic C-H acylation using nickel photoredox catalysis. Carboxylic acids were used as acyl surrogates which resulted in the formation of alpha-arylketones. After reaction optimization, an extensive substrate scope is presented, including applications for late stage functionalizations. A gram scale synthesis demonstrates the ability for upscaling and the usefulness of the method for the parallel synthesis of drug analogues is revealed. Mechanistic experiments were performed to support the proposed mechanism in which a bromine radical is responsible for the C-H activation by HAT. Finally, an expansion to the synthesis of related alpha-aryl esters is reported.

This is a very interesting work which introduces a very elegant, mild and straightforward method for transforming readily available carboxylic acid feedstock molecules and simple alkylbenzenes into more valuable alpha-arylketones and in extension alpha-arylesters. The extensive scope and late stage functionalization reactions demonstrate that this method is robust and versatile. In my opinion, this elegant new photocatalytic method is a great contribution for publication in Nature Communications.

Just a few minor points:

-Despite the extensive scope, it is noticeable that only one example is shown in which a benzene is replaced with a heteroaromatic system (thiophene). Did the authors try indole or protected indoles?

-Regarding the variation of alkylarenes it is also noticeable that the ortho-substituted bromine provides the product in low yields (37%). The system seems to be quite sensitive to steric effects. This should be stated somewhere.

Overall, I suggest that the authors give the reader a better idea of the scope AND limitations of the method. It is not only important to feature substrates which work well, but also substrates which don't. This won't diminish the accomplishments of the authors but will give the reader a better understanding.

Point-by-point response to the reviewers' comments

Reviewer 1:

“This reviewer has some mixed feelings about the appropriateness of this submission for publication in Nature Communications. On the one hand, the method developed looks very robust and versatile, and a high level of rigor and attention to detail is exhibited. On the other hand, the method is somewhat derivative of previous research, and therefore conceptually not much is added to the literature other than the report of a facile route to a class of molecules that are, admittedly, important....Overall a very solid contribution, but perhaps lacking a bit in novelty and scholarship.”

General response: We thank this reviewer for insightful comments and suggestions, which have improved the quality of the manuscript. This method provides a strategic alternative to address the challenging stereocontrol in benzylic C(sp³)-H functionalization and Ni/photoredox dual catalysis. To our knowledge, it would be the first example of enantioselective benzylic C-H acylation. We have addressed the concerns of this reviewer. Please see the details as below:

Comment 1: “In places there is a need for an English-speaking reader to correct for idiomatic English.”

Our response: The revised manuscript has been edited by a native English speaker. The changes have been highlighted in yellow.

Comment 2: “There is not a great deal of attention paid to previous asymmetric processes through photoredox catalysis. The authors are referred to a recent review: Angew. Chem. Int. Ed. 2021, 60, 1714 – 1726.”

Our response: We have cited this paper as reference 35 in the revised manuscript.

Comment 3: “There is also short-shrift paid to previous acylation reactions conducted by in situ activation of carboxylic acids. See, for example, Org. Lett. 2017, 19, 3612–3615 and references therein. There are several other examples since this, but this is one of the earliest examples using dimethyl dicarbonate.”

Our response: We have added a reference to this early work of in situ activation of carboxylic acids (reference 59).

Comment 4: “It is not clear what differentiates acetate and esters (lines 94-95). They are both esters.”

Our response: This error has been corrected in the revised manuscript.

Comment 5: “In lines 35 and 36, the compounds are not boranes, but rather boronates.”

Our response: “borane” has been changed to “boronate ester” in the revised manuscript.

Comment 6: “The chemistry in Figure 4B with Flurbiprofen is not well illustrated – it is not clear from the graphics exactly what is being done. The wrong bond disconnects are highlighted and the colors are not consistent with those used in the rest of the manuscript. This must be corrected for clarity.”

Our response: To clarify this, the colors for the bond disconnects have been adjusted in Figure 4.

Comment 7: “In Figure 5, it is surprising that addition to the alkene was competitive with capture of the radical by Ni complex, a process that is essentially barrierless. Perhaps this is just the effect of concentration in that there is a large excess of the alkene in contrast to the catalytic Ni complexes.”

Our response: Yes, they are two competitive pathways. If decrease the loading of alkene, both the products of Giese addition and acylation could be formed simultaneously. In addition, the newly formed phenylsulfonic acid from the addition to the alkene is likely to make the in-situ formed anhydride to decompose or inhibit its formation.

Comment 8: “A glaring omission is that the proposed mechanism does not address the source of asymmetric induction. Thus, the enantioselectivity could result from the irreversible diastereoselective formation of the Ni complex or a dynamic kinetic transformation with reversible addition of the radical to the nickel complex. There are calculations from other groups that address this issue, but no discussion is evident in this contribution.”

Our response: To explain this, we have added a sentence to the text of the revised manuscript:

“A recent computational study of nickel-catalyzed cross-coupling of photoredox-generated benzylic radicals suggested that reductive elimination is the stereochemistry-determining step.”

We have also added a reference (*JACS*, **2015**, *137*, 4896) to this mechanistic study (reference 55). Moreover, the addition of benzylic radical to the nickel complex in the proposed mechanism has been changed to a reversible process.

Comment 9: “It is not clear from the experimental description how the reactions were kept at 10 °C during irradiation. This should be explicitly documented.”

Our response: To clarify this, a picture of exemplary reaction setup (Supplementary Figure 1) has been added to the Supplementary Information (Page S-5).

Comment 10: “There are no melting points reported for any of the solids.”

Our response: All the melting points have been added to the solid products in the Supplementary Information.

Reviewer 2:

“In this manuscript, Huan and coworkers report a photoredox/Ni/HAT promoted asymmetric acylation of benzylic C-H bonds to access α -aryl ketones with broad substrate scope and functional group tolerance. While the photoredox/Ni catalytic platform has been well established, the development of asymmetric transformations remains less explored. Although this paper is a logical extension of their previously reported work (ref. 37) using alkyl arenes instead of amide as C-H coupling partner under very similar conditions, the synthesis of chiral α -aryl ketone in the manuscript exhibits meaningful complementarity (simple use of acids as acyl source and alkyl arenes as prochiral C-H nucleophiles) compared with previous enantioselective approaches by transition-metal catalysis, which is especially well demonstrated in Fig. 3. Overall, this is a well done, very nice study that should be accepted after some revisions noted below.”

General response: We thank this reviewer for supporting the publication of this work in *Nature Communications*. We also thank this reviewer for insightful comments and suggestions, which have improved the quality of the manuscript. We have addressed the concerns of this reviewer as below:

Comment 1: “The authors demonstrate aromatic and aliphatic (primary) acids as partners. Some comment would be useful with respect to the use of secondary and tertiary acids?”

Our response: The secondary acids led to low efficiency under the current reaction conditions (for isobutyric acid: 33% yield, 84% ee; for cyclohexanecarboxylic acid: 36% yield, 71% ee). Unfortunately, tertiary acids such as pivalic acid suppresses the reactivity of acylation completely. We have added these results in the Supplementary table 1, and also added a comment in the text to the scope limitation:

“Under the current reaction conditions, the sterically hindered coupling partners such as the α -branched carboxylic acids and *ortho*-substituted alkylarene (**33**) led to low efficiency or no product formation (also see the Supplementary Table 1)”

Comment 2: “Why do aromatic acids dramatically lower reactivity and enantioselectivity? A comment would be useful.”

Our response: To clarify this, we have added a comment in the text of the revised manuscript:

“The corresponding methyl carboxylate was a significant side product for the cross-coupling of aromatic carboxylic acids.”

Comment 3: “Argument of line 106 is specious. The capability of a method is not justified by whether the product has been previously prepared or not. Clearly, if the methyl substrate is known, one can make the ethyl derivative and claim new chemical space. This should be deleted. Also, in line 106, ‘figure 3’ should be ‘figure 2’.”

Our response: The following sentence has been deleted in the text of the revised manuscript:

“Notably, over 90% of the enantioenriched compounds illustrated in Figure 3 have never been prepared before, thereby illustrating the capability of this method to access new chemical space.”

Comment 4: “The colors in Fig. 3 B make it confusing for the disconnections of coupling partners. Sometimes the blue is acid, sometimes not, sometimes it includes the carbonyl, sometimes not. This should be fixed.”

Our response: To clarify this, the colors for the bond disconnects have been adjusted in Figure 4.

Comment 5: “A recent publication about partial activation of alcohols in cross electrophile coupling reveals an intriguing concept (JACS2021/143/513). A simple experiment can verify whether the in-situ acid activation follows this kinetics.”

Our response: We have monitored the anhydride formation by the GC and ¹H-NMR analysis under the standard conditions without blue LED irradiation. As show below, it was found that the carboxylic acid was almost fully converted into mixed anhydride (49%) and symmetrical anhydride (25%) within 30 min. The in-situ acid activation does not follow the partial activation mode. The use of symmetrical anhydride (76% yield, 95% ee, shown in Table 1, entry 9) or mixed anhydride (73% yield, 95% ee instead of the combination of carboxylic acid, DMDC, and NH₄Cl led to comparable results.

Comment 6: “After acylation, the benzylic position in the product is deactivated. In principle, less activated C-H precursors should also work.”

Our response: The weakness of the H–Br bond (BDE = 88 kcal/mol) provide a mild thermodynamic driving force for the C–H cleavage, and the bromine radicals have limited ability to engage stronger C–H bonds (normally BDE >90 kcal/mol) in these reactions (Chem. Rev. **2017**, *117*, 8622.).

Comment 7: “The paper (JACS2015/137/4896) discusses stereoconvergence in Photoredox/Ni-catalyzed cross coupling involving benzylic radical capture and should be cited.”

Our response: To explain this, we have added a sentence to the text of the revised manuscript:

“A recent computational study of nickel-catalyzed cross-coupling of photoredox-generated benzylic radicals suggested that reductive elimination is the stereochemistry-determining step.”

We have also added a reference (*JACS*, **2015**, *137*, 4896) to this mechanistic study (reference 55). Moreover, the addition of benzylic radical to the nickel complex in the proposed mechanism has been changed to reversible process.

Reviewer 3:

“The manuscript by Huo and coworkers reports a new method for the asymmetric benzylic C-H acylation using nickel photoredox catalysis. Carboxylic acids were used as acyl surrogates which resulted in the formation of alpha-arylketones. After reaction optimization, an extensive substrate scope is presented, including applications for late stage functionalizations. A gram scale synthesis demonstrates the ability for upscaling and the usefulness of the method for the parallel synthesis of drug analogues is revealed. Mechanistic experiments were performed to support the proposed mechanism in which a bromine radical is responsible for the C-H activation by HAT. Finally, an expansion to the synthesis of related alpha-aryl esters is reported.

This is a very interesting work which introduces a very elegant, mild and straightforward method for transforming readily available carboxylic acid feedstock molecules and simple alkylbenzenes into more valuable alpha-arylketones and in extension alpha-arylesters. The extensive scope and late stage functionalization reactions demonstrate that this method is robust and versatile. In my opinion, this elegant new photocatalytic method is a great contribution for publication in *Nature Communications*.”

General response: We thank this reviewer for supporting the publication of this work in *Nature Communications*. We also thank this reviewer for insightful comments and suggestions, which have improved the quality of the manuscript. We have addressed the concerns of this reviewer as below:

Comment 1: “Despite the extensive scope, it is noticeable that only one example is shown in which a benzene is replaced with a heteroaromatic system (thiophene). Did the authors try indole or protected indoles?”

Our response: We have added the below two examples in the Figure 2 (compound **46**) and the Supplementary Table 1 (entry 2), respectively.

73% yield, 95% ee

42% yield, 92% ee

Comment 2: “Regarding the variation of alkylarenes it is also noticeable that the *ortho*-substituted bromine provides the product in low yields (37%). The system seems to be quite sensitive to steric effects. This should be stated somewhere.”

Our response: We have added a comment in the text to the scope limitation:

“Under the current reaction conditions, the sterically hindered coupling partners such as the α -branched carboxylic acids and *ortho*-substituted alkylarene (**33**) led to low efficiency or no product formation (also see the Supplementary Table 1)”

Comment 3: “Overall, I suggest that the authors give the reader a better idea of the scope AND limitations of the method. It is not only important to feature substrates which work well, but also substrates which don't. This won't diminish the accomplishments of the authors but will give the reader a better understanding.”

Our response: To give the reader a better view of the scope and its limitations, we have added the related results in the Supplementary table 1.

REVIEWERS' COMMENTS

Reviewer #1 (Remarks to the Author):

The authors have done a reasonable job of addressing the previous comments.

Reviewer #2 (Remarks to the Author):

My comments from the first version of this manuscript have been satisfactorily addressed for the most part (the scientific content has been - some of the stylistic questions have not but I leave it to the authors' and editor's discretion). I was supportive the first time and I view this as now being perfectly suitable for publication. It's a nice piece of work.

Reviewer #3 (Remarks to the Author):

The authors addressed all my previous comments in a satisfactory fashion. This excellent manuscript is ready to go.

Point-by-point response to the reviewers' comments

Reviewer 1:

“The authors have done a reasonable job of addressing the previous comments.”

Our response: We thank this reviewer for supporting the publication of this work in *Nature Communications*. We also thank this reviewer for previous insightful comments and suggestions, which have improved the quality of the manuscript.

Reviewer 2:

“My comments from the first version of this manuscript have been satisfactorily addressed for the most part (the scientific content has been - some of the stylistic questions have not but I leave it to the authors' and editor's discretion). I was supportive the first time and I view this as now being perfectly suitable for publication. It's a nice piece of work.”

Our response: We thank this reviewer for supporting the publication of this work in *Nature Communications*. We also thank this reviewer for previous insightful comments and suggestions, which have improved the quality of the manuscript. We have addressed the editorial stylistic requests in the final version of manuscript and the revised author checklist.

Reviewer 3:

“The authors addressed all my previous comments in a satisfactory fashion. This excellent manuscript is ready to go.”

Our response: We thank this reviewer for supporting the publication of this work in *Nature Communications*. We also thank this reviewer for previous insightful comments and suggestions, which have improved the quality of the manuscript.